# Simultaneously Enhancing the Efficiency and Stability of Perovskite Solar Cells by Using P3HT/PEDOT:PSS as a Double Hole Transport Layer

**DOI:** 10.3390/nano14181476

**Published:** 2024-09-11

**Authors:** Xiude Yang, Minghao Luo, Qianqian Zhang, Haishen Huang, Yanqing Yao, Yuanlin Yang, Ying Li, Wan Cheng, Ping Li

**Affiliations:** 1School of Physics and Electronic Science, Zunyi Normal College, Zunyi 563006, China; yxiude@163.com (X.Y.);; 2College of Physics and Electronic Engineering, Chongqing Normal University, Chongqing 401331, China

**Keywords:** perovskite solar cells, P3HT, PEDOT:PSS, stability

## Abstract

The stability issue of perovskite solar cells (PSCs) has long been of concern to researchers. Poly (3,4-ethylenedioxythiophene) polystyrene sulfonate (PEDOT:PSS) is commonly used as a hole transport layer (HTL) in the inverted PSCs to achieve efficient and stable performance. However, PEDOT:PSS can corrode ITO, affecting device efficiency. Moreover, the hydrophilic nature of PEDOT:PSS compromises device stability. In this work, Poly (3-hexylthiophene-2,5-diyl) (P3HT), known for its good hydrophobicity, was used to modify the surface of PEDOT:PSS, reducing its water absorption and thereby enhancing the efficiency and stability of PSCs. The results reveal that incorporating P3HT effectively enhances the hydrophobicity of PEDOT:PSS. Furthermore, it fosters the development of large-grain perovskite film on the PEDOT:PSS/P3HT bilayer. This enhancement leads to a power conversion efficiency (PCE) of 19.78% for PSCs, with an increase by 16% than that of reference cells (17.04% of PCE). Following a duration of 1000 h, the PCE for the device modified with P3HT remains above 90%, while the PCE of the reference device is below 70%. These findings suggest that using P3HT in conjunction with PEDOT:PSS as a bilayer HTL can concurrently and proficiently improve the efficiency and stability of PSCs.

## 1. Introduction

Organic–inorganic halide perovskite solar cells (PSCs) have garnered substantial attention owing to their exceptional power conversion efficiency (PCE), capability for low-temperature processing, and low-effective manufacturing [1,2,3,4]. Especially, PCE of the device has increased from the initial 3.8% to more than 25.8% [5,6]. However, PSCs have poor stability because factors including light, temperature, oxygen, and water in the air can cause perovskite to decompose [7,8]. Bert Conings et al. [9] theoretically calculated the structure of perovskite and found significant decomposition while annealing at 85 °C in an inert atmosphere. Kamat et al. [10] showed that the interaction between methylammonium triiodoplumbate (CH_3_NH_3_PbI_3_) and water vapor can lead to the formation of hydrate products similar to (CH_3_NH_3_) 4PbI_6_·2H_2_O. This results in significant changes in the crystal structure of the perovskite material and reduces light absorption. Therefore, it is very important to prevent decomposition of perovskite.

A good hole transport layer material can help to achieve effective energy level matching between the perovskite active layer and electrode. This improves hole extraction and inhibits electron–hole recombination at the interface between the perovskite active layer and electrode, thereby reducing carrier loss [11,12]. Poly (3,4-ethylenedioxythiophene) polystyrene sulfonate (PEDOT:PSS) is extensively employed in the inverted structure of organic–inorganic hybrid PSCs. This is attributed to its numerous advantages, such as ease of low-temperature processing, high transmittance, flexibility, electrical conductivity, and suitable hole mobility [13,14,15]. However, PEDOT:PSS will lead to indium tin oxide (ITO) corrosion, perovskite crystal destruction, and low open-circuit voltage (V_oc_) values because of its strong acidity, water absorption, weak conductivity, and mismatch energy level between electrode and perovskite energy band, thereby affecting the performance of PSCs [16,17,18,19]. To overcome the deficiencies of PEDOT:PSS, many methods were used to modify PEDOT:PSS. Ligang Xu et al. [20] proposed that implementing low-temperature annealing after the incorporation of copper(II) thiocyanate into PEDOT:PSS could augment the mean size of perovskite crystals and bolster the stability of inverted PSCs. Yi Chun Chin et al. [2] proposed that doping sodium hydroxide (NaOH) aqueous solution in PEDOT:PSS can reduce the charge recombination loss at PEDOT:PSS/perovskite interface, thereby enhancing the overall performance of the device. Clearly, modifying PEDOT: PSS can not only boost the performance of the device but also enhance its stability. PEDOT:PSS with hydrophilicity has strong water absorption, thereby reducing the stability of the device. Many studies show that using hydrophobic polymers can significantly enhance the efficiency of perovskite solar cells with conventional PEDOT:PSS as the hole transport layer [21,22,23,24,25,26]; see Appendix A. These improvements are mainly attributed to the protective effect of hydrophobic polymers on the perovskite layer and the modification of PEDOT:PSS, which reduces interface defects and the impact of moisture. Poly(3-hexylthiophene-2,5-diyl) (P3HT), characterized by its pronounced hydrophobic nature, is extensively utilized in organic solar cells as the donor material within the active layer, enabling the fabrication of polymer-based solar cells with enhanced stability [27]. P3HT exhibits a high hole mobility up to 0.1 cm^2^ V^−1^ s^−1^ [28,29], which is recognized as an alternative polymeric hole-transport material. It boasts superior optoelectronic properties, low cost, and ease of fabrication, making it a highly attractive option for use in electronic devices [30]. The use of P3HT as a top-hole transport layer (HTL) was explored in a previous report to improve the efficiency and stability of the PSCs [31]. But the preparation of PSCs using P3HT as the bottom HTL has rarely been reported because of difficulty in forming a uniform perovskite film on P3HT HTL.

In this work, P3HT was incorporated between the PEDOT:PSS and perovskite active layer, which effectively prevented the corrosion of the perovskite active layer caused by the strong acidity of PEDOT:PSS. Additionally, owing to the excellent hydrophobicity of P3HT, it is anticipated to mitigate the impact of the water absorption by PEDOT:PSS on the perovskite layer, thereby enhancing the stability of the PSCs. Consequently, the influence of introducing P3HT between the PEDOT:PSS and perovskite active layer on the performance and stability of the PSCs is examined in extensive detail.

## 2. Experimental Section

### 2.1. Materials

Silver (Ag), bathocuproine (BCP), methyl ammonium iodide (MAI), formamidine ammonium iodide (FAI), dimethyl sulfoxide (DMSO), lead chloride (PbCl_2_), [66]-phenyl C61 methyl butyrate (PCBM), poly (3-hexylthiophene-2,5-diyl) (P3HT), N′-dimethylformamide (DMF), PEDOT:PSS (4083) were procured from Xi’an Polymer Light Technology Corp. (Xi’an, China) Ethanol and chlorobenzene (CB) were obtained from Beijing Bailingwei Technology Corp. (Bei’jing, China). Moreover, lead (II) iodide (PbI_2_) was acquired from Liaoning Youxuan New Energy Technology Corp. (Dalian, China)

### 2.2. Device Preparation

PSCs with the structure of ITO/PEDOT:PSS/X-P3HT/perovskite/PCBM/BCP/Ag were fabricated, where X is P3HT concentration of 0.01, 0.02, 0.03, and 0.04 mg/mL in CB, respectively. According to the X difference, these PSCs were marked as pure PEDOT:PSS, 0.01-P3HT/PEDOT:PSS, 0.02-P3HT/PEDOT:PSS, 0.03-P3HT/PEDOT:PSS, and 0.04-P3HT/PEDOT:PSS devices, respectively. First, ITO/glass substrates underwent a thorough cleaning process. They were ultrasonically scrubbed in a sequence of solutions: deionized water, detergent, deionized water again, acetone, and ethyl alcohol, with each step lasting for 15 min. After that, the substrates were dried using a flow of nitrogen gas. Subsequently, a plasma treatment was administered for a duration of 2 min to further purify the surface. After these preparatory steps, a thin film of PEDOT:PSS was precisely deposited onto the substrates. Filter PEDOT:PSS, and then dilute the filtered PEDOT:PSS with deionized water in a 1:4 ratio. PEDOT:PSS HTLs were uniformly spin-coated onto the ITO glass substrates at a rotational speed of 4000 rpm for a duration of 40 s. Following the spin-coating process, the PEDOT:PSS-coated samples underwent a thermal treatment at 130 °C for a period of 20 min. Subsequent to this heating step, all samples were carefully transferred into a nitrogen-filled glove box to proceed with the subsequent device fabrication steps. A certain amount of P3HT was dissolved in CB for filtration and dilution. The dilution concentrations were 0.01, 0.02, 0.03, and 0.04 mg/mL, respectively. A 40 µL solution of P3HT solutions, varying in concentrations, was spin-coated onto the samples pre-coated with PEDOT:PSS at a rotational speed of 6000 rpm for 60 s. After spin-coating, the samples were left to stand for an additional 30 min to ensure uniform coating. The perovskite precursor solution was meticulously prepared by combining 200.3 mg of MAI, 24 mg of FAI, 580.9 mg of PbI_2_, and 38.9 mg of PbCl_2_ in a mixture of 1 mL anhydrous DMF and DMSO co-solvent, maintaining a volume ratio of 9:1. The formation of the perovskite layer was achieved by spin-coating the precursor solution onto the PEDOT:PSS-coated ITO substrate, which had various concentrations of P3HT applied to them, at a rate of 4000 rpm for 30 s. During the initiation of the perovskite film’s spin-coating process, after 12 s, 200 mL of anhydrous CB was introduced dropwise onto the spinning sample. Following this, the perovskite layer-coated samples underwent a two-step annealing process: initially at 50 °C for 2 min, then at 85 °C for an additional 30 min. For the formation of the electron transport layer (ETL), a PCBM solution (20 mg/mL in CB) devoid of any additive was spin-coated atop the perovskite layer at 5000 rpm for 35 s. Subsequently, BCP dissolved in IPA was applied onto the PCBM layers through spin-coating at 2000 rpm for 30 s. In the final step, an 80 nm thick metal silver electrode was thermally evaporated onto the solar cell under high vacuum using a shadow mask. Each fabricated device encompassed an active area of 0.09 cm^2^.

### 2.3. Characterization

The contact angle of the film was measured using a microscopic contact angle meter (CA, JC2000D6) (Shanghai, China). The surface morphology of the perovskite film was characterized using an atomic force microscope (AFM, model CSPM5500) ((Guangzhou, China)) and a scanning electron microscope (SEM, JSM-6700F) (Shanghai, China). Additionally, the crystal structure of the film was analyzed using an X-ray diffractometer (XRD-7000) (Shenzhen, China). The absorption characteristics of the film were analyzed using a Varian Cary 5000 UV-Vis-NIR spectrometer (Shanghai, China). The photovoltaic performance of the PSCs, which utilized PEDOT:PSS films as the HTL, was evaluated in a glovebox. This evaluation was conducted using a computer-programmed Keithley 2400 source/m coupled with a Newport solar simulator. The simulator reproduced AM 1.5 sunlight conditions at a light intensity of 100 mW·cm^2^, approximating one sun illumination. The encapsulated PSCs were subjected to a stability test, where they were stored in an environment with a temperature of 25 °C and humidity levels ranging from 30% to 35%. The testing was conducted under ambient air conditions.

## 3. Results and Discussion

### 3.1. Hydrophobicity

To mitigate the impact of water absorption by PEDOT:PSS on the perovskite film, a layer of P3HT was incorporated to modify PEDOT:PSS. The contact angles of pure PEDOT:PSS and PEDOT:PSS with various concentrations of P3HT were measured, as depicted in Figure 1. It is evident that the untreated PEDOT:PSS film exhibits superior hydrophilic properties (Figure 1a), while the introduction of P3HT greatly reduces the hydrophilicity of PEDOT:PSS (Figure 1b). When using water and perovskite solutions as droplets for contact angle measurement, it was found that both the contact angle of water and the perovskite solution showed different increases, which indicated that the introduction of P3HT improved the hydrophobicity of the PEDOT:PSS/P3HT composite film. Figure 1c shows that the contact angle between water and perovskite solution increases with the increase in P3HT concentration, but the change in trend of the perovskite solution is not as fast as that of water, which indicates that the introduction of P3HT can solve the problem of water absorption to some extent. However, if the hydrophobicity is too good, it will adversely affect the formation of the perovskite films. In this study, it has been found that it is very difficult to form uniform perovskite films when the P3HT concentration is higher than 0.05 mg/mL. Therefore, various concentrations of P3HT, including 0.01, 0.02, 0.03, and 0.04 mg/mL, were studied in the below work. The contact angle measurements of the optimized film (0.03 mg/mL) with water and perovskite solution are 56 and 24 degrees, respectively.

### 3.2. Morphology

To examine the impact of different concentrations of P3HT-modified PEDOT:PSS films on the morphology of perovskite films, the SEM morphology of ITO/PEDOT:PSS/P3HT/Perovskite and ITO/PEDOT:PSS/Perovskite half-device structures is measured and shown in Figure 2. After conducting measurement and calculation, the average grain size of the perovskite film made with pure PEDOT:PSS is 369.36 nm. The incorporation of P3HT results in an enlargement of the perovskite film’s average grain size. When P3HT concentration is 0.03 mg/mL, the overall crystallization of the perovskite film is optimal, with an average grain diameter of 476.67 nm. The increase of grain size is attributed to the introduction of P3HT, which reduces the PEDOT:PSS surface wetting—thereby promoting, to some extent, the formation of high-quality perovskite films and the growth of larger grains [32].

The AFM morphology of the perovskite film on pure PEDOT:PSS and various concentrations of P3HT are shown in Figure 3. Notably, the roughness of the perovskite film on pure PEDOT:PSS is 23.09 nm. The roughness of the perovskite film decreases as the concentration of P3HT increases. The minimum roughness of 19.45 nm is achieved for perovskite film on 0.03-P3HT/PEDOT:PSS HTL. The decrease in roughness can be attributed to the enlargement of perovskite grain size. The findings further suggest that the introduction of P3HT has substantially enhanced the crystallinity of perovskite films, which could potentially improve light absorption and scattering.

### 3.3. UV-Vis Absorption Spectrum

To examine the impact of varying P3HT concentrations on the optical absorption of perovskite thin films, UV-Vis spectra were obtained for ITO/PEDOT:PSS/P3HT/Perovskite and ITO/PEDOT:PSS/Perovskite half-device configurations, as depicted in Figure 4. It is evident from the data that all perovskite thin films exhibit nearly uniform absorption across wavelengths of 600~850 nm and 300~490 nm. However, the absorption capability of perovskite film incorporating P3HT is markedly superior to that of the perovskite film with pure PEDOT:PSS within the 490~600 nm wavelength range. The enhancement in absorption may be due to the contribution of P3HT absorption [33,34]; see Appendix A.

### 3.4. Crystal Structure

To further demonstrate the introduction of P3HT facilitates the crystallization of perovskite, the XRD images of ITO/PEDOT:PSS/P3HT/Perovskite and ITO/PEDOT:PSS/Perovskite half-devices were carried out and shown in Figure 5. According to the analysis from Figure 5, the (100) characteristic peak intensity of the perovskite film, fabricated on 0.03-P3HT/PEDOT:PSS HTL, is markedly superior to that of the perovskite film fabricated on pure PEDOT:PSS. While the PbI_2_ peak of the perovskite film combined with P3HT is lower than that of the perovskite film with pure PEDOT:PSS. These results indicate that incorporating P3HT can significantly enhance the crystallinity of perovskite and inhibit the decomposition of the perovskite duo to hinder its absorption of water (Figure 1).

### 3.5. Efficiency and Stablity

The structures as well as the energy band scheme of the fabricated PSC devices were shown in Figure 6a,b. The I-V characteristics of PSCs incorporating various HTLs were evaluated under standard testing conditions of 100 mW·cm^−2^ using an AM 1.5G solar simulator. Figure 6c shows the performance of PSCs fabricated on pure PEDOT:PSS and 0.03-P3HT/PEDOT:PSS. All devices demonstrated satisfactory performance, as summarized in Table 1. The data within parentheses indicates the optimal performance of the inverted solar cells. With pure PEDOT:PSS HTL, the device gives a PCE of 16.49 ± 0.6%, with the J_sc_ of 20.95 ± 0.91 mA·cm^−2^, V_oc_ of 1.00 ± 0.01 V, and the FF of 76.11 ± 1.82%. Inserting P3HT as a HTL between PEDOT:PSS and the perovskite active layer significantly enhances the performance of the PSCs. The J_sc_ rises from 20.95 ± 0.91 to 22.94 ± 0.76 mA·cm^−2^ as the concentration of P3HT increases from 0 to 0.03 mg/mL. However, it gradually declines to 21.85 ± 0.99 mA·cm^−2^ upon further increasing the P3HT concentration to 0.04 mg/mL. The highest PCE value of 19.78% is achieved for the PSCs with a 0.03-P3HT/PEDOT:PSS bilayer HTL, which is 16% higher than that of the reference device (17.04% of PCE). The enhanced performance with P3HT/PEDOT:PSS as HTL is mainly due to improved charge transportation and reduction of nonradiative charge recombination. The EQEs of PSCs with pure PEDOT:PSS and 0.03-P3HT/PEDOT:PSS HTLs were compared to investigate the mechanism behind the J_sc_ enhancement when P3HT was inserted between PEDOT:PSS and the perovskite active layer, as shown in Figure 6d. The EQEs of PSCs with 0.03-P3HT/PEDOT:PSS HTL are higher than those of PSCs with pure PEDOT:PSS. This outcome aligns with the I-V curve measurement. The increase in EQEs could be ascribed to the enhanced charge transport capability. The improvement in charge transport capability is attributing to the effective energy level arrangement of P3HT, which provides a platform for hole transport and facilitates the transport of holes from the perovskite layer to the PEDOT:PSS layer (Figure 6b).

A high-conductivity interface layer is beneficial to charge transport and extraction [35]. The comparison of conductivity between pure PEDOT:PSS and 0.03-P3HT/PEDOT:PSS HTLs is depicted in Figure 7a. The conductivity of the device with 0.03-P3HT/PEDOT:PSS HTL was enhanced by one order of magnitude, from 7.98 × 10^−6^ s.cm^−1^ to 1.31 × 10^−5^ s.cm^−1^, compared to the device using pure PEDOT:PSS HTL. The conductivity results are consistent with the performance of the PSCs using pure PEDOT:PSS and 0.03-P3HT/PEDOT:PSS HTLs (Figure 6a), indicating the increased charge transport ability by P3HT incorporation. To explore the effect of the introduction of P3HT on interface charge recombination, the photoluminescence (PL) was measured, focusing on those with a perovskite film deposited on both pure PEDOT:PSS and 0.03-P3HT/PEDOT:PSS HTLs. The PL intensity of the perovskite films, when formed on the 0.03-P3HT/PEDOT:PSS HTL, is significantly lower compared to those deposited on a pure PEDOT:PSS HTL, as shown in Figure 7b. This result indicates that nonradiative charge recombination was effectively reduced through the insertion of P3HT at the interface between the perovskite film and PEDOT:PSS HTL. The reduction of nonradiative charge recombination is attributed to carrier extraction at the interface between the perovskite film and PEDOT:PSS HTL. The co-solvent treatment results in faster charge transfer and extraction, leading to a reduction in the PL intensity [36,37,38,39].

To investigate the effect of P3HT introduction on stability of PSCs, the performance of PSCs with pure PEDOT:PSS and 0.03-P3HT/PEDOT:PSS HTLs stored under environmental conditions of 25 °C temperature and 30~35% humidity was tested by long-term tracking as shown in Figure 8. All the devices were stored in the dark and only illuminated when testing I-V. The efficiency of PSCs, which use 0.03-P3HT/PEDOT:PSS HTLs, is observed to remain above 90% of its initial value even after 1000 h of measurement in an ambient air environment, while the efficiency of PSCs with pure PEDOT:PSS decreases to below 70% of the initial efficiency. The findings suggest that the introduction of P3HT into PEDOT:PSS helps to reduce water absorption by and effectively inhibits the degradation of perovskite films, which ultimately increases the lifetime of the device and significantly improves its long-term stability.

## 4. Conclusions

This work investigates the effect of inserting P3HT between PEDOT:PSS and the perovskite active layer on the performance and stability of PSCs. It can be found that the introduction of P3HT effectively improves the hydrophobicity of PEDOT:PSS film and promotes the crystallization of the perovskite active layer. Notably, when the P3HT concentration reaches 0.03 mg/mL, the perovskite film exhibits optimal crystallinity, characterized by an average grain size of 476.67 nm. Furthermore, XRD test results prove that the introduction of P3HT enhances the crystallization of perovskite film. Moreover, the introduction of P3HT can effectively facilitate charge transport and extract at the interface between PEDOT:PSS and the perovskite active layer, as well as reducing the nonradiative charge recombination. The optimal PCE of 19.78% for PSC with P3HT/PEDOT:PSS as HTL is achieved, which is about 16% higher than that of PSC with pure PEDOT:PSS as HTL. The stability test of the PSCs shows that PSC-based P3HT still maintains above 90% of the initial efficiency after 1000 h, while the efficiency of reference PSCs decreases to below 70% of the initial efficiency. In summary, the introduction of P3HT between PEDOT:PSS and the perovskite active layer can simultaneously and effectively improve the efficiency and stability of PSCs.

## Figures and Tables

**Figure 1 nanomaterials-14-01476-f001:**
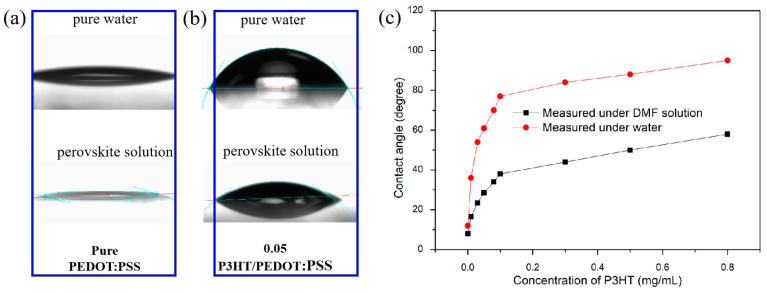
Contact angle image of PEDOT: PSS surface modified (**a**) without and (**b**) with P3HT using deionized water and perovskite solution (DMF solution). (**c**) Change of contact angle of PEDOT:PSS modified with different P3HT concentrations.

**Figure 2 nanomaterials-14-01476-f002:**
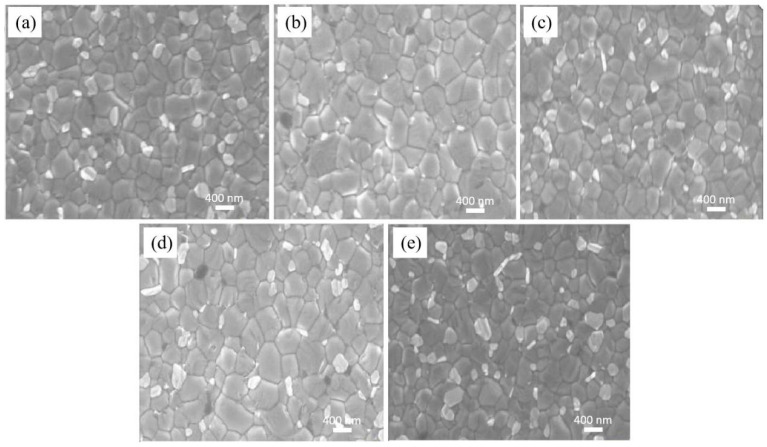
Analysis of the grain size of the perovskite films on PEDOT:PSS without and with varying P3HT concentrations. (**a**) pure PEDOT:PSS, (**b**) 0.01 mg/mL, (**c**) 0.02 mg/mL, (**d**) 0.03 mg/mL, and (**e**) 0.04 mg/mL.

**Figure 3 nanomaterials-14-01476-f003:**
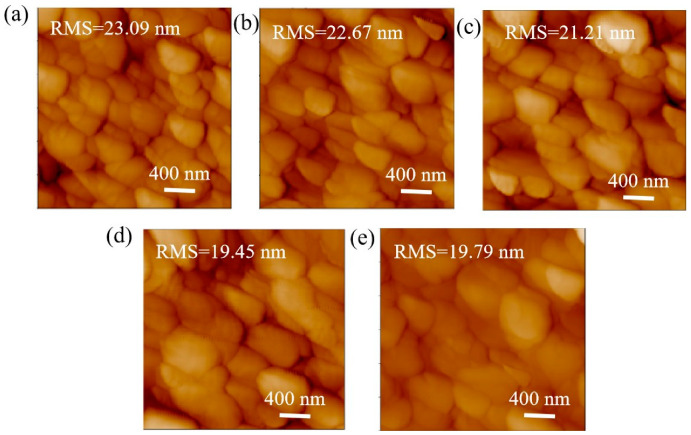
AFM images of perovskite films on PEDOT:PSS without and with varying P3HT concentrations. (**a**) pure PEDOT:PSS, (**b**) 0.01 mg/mL, (**c**) 0.02 mg/mL, (**d**) 0.03 mg/mL, and (**e**) 0.04 mg/mL.

**Figure 4 nanomaterials-14-01476-f004:**
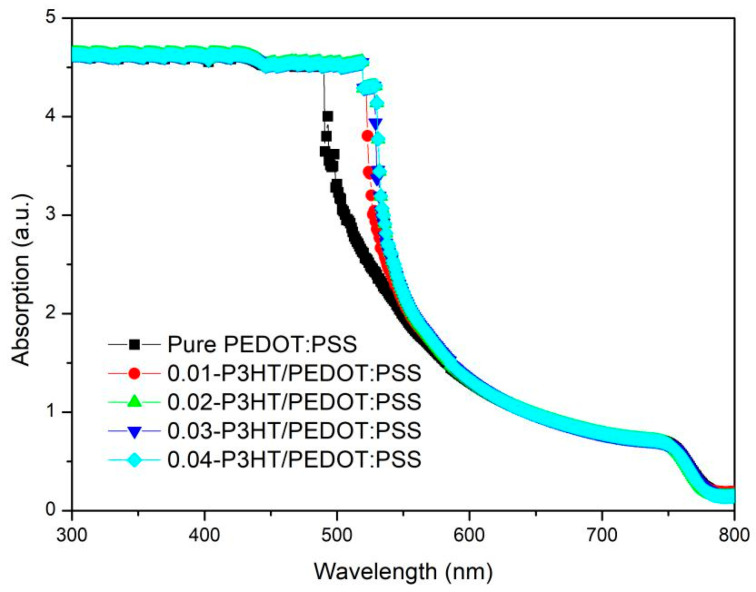
UV-Vis absorption spectra of perovskite films fabricated on the pure PEDOT:PSS and various P3HT concentrations.

**Figure 5 nanomaterials-14-01476-f005:**
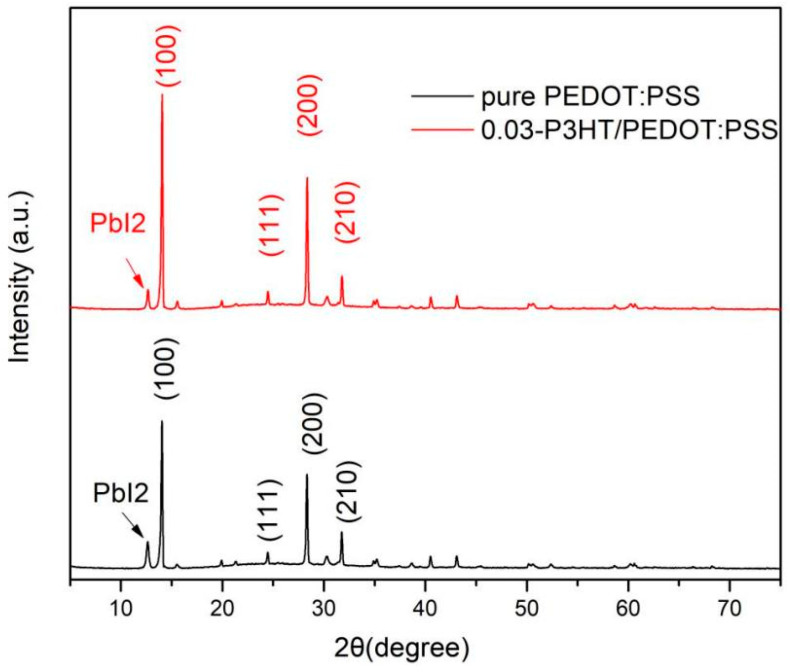
XRD images of the perovskite film fabricated on pure PEDOT:PSS and 0.03-P3HT/PEDOT:PSS.

**Figure 6 nanomaterials-14-01476-f006:**
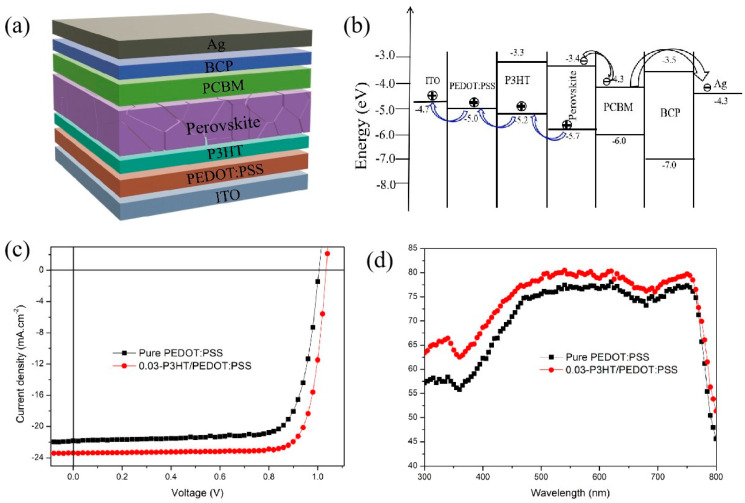
(**a**) Structures and (**b**) energy band scheme of fabricated PSCs; (**c**) I-V curves; and (**d**) EQE of the PSCs based on pure PEDOT:PSS HTL and 0.03-P3HT/PEDOT:PSS HTL.

**Figure 7 nanomaterials-14-01476-f007:**
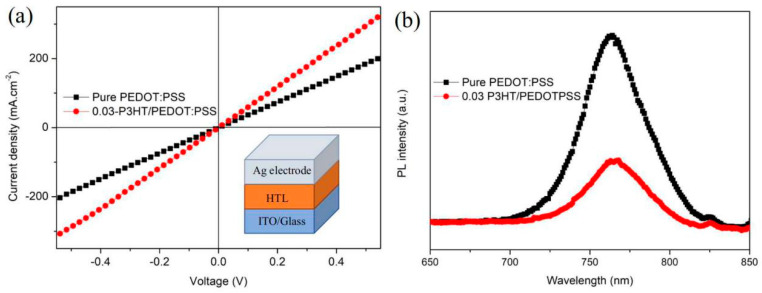
(**a**) Conductivity of the PSCs using the pure PEDOT:PSS HTL and the 0.03-P3HT/PEDOT:PSS HTL. (**b**) PL of perovskite film fabricated on the pure PEDOT:PSS HTL and the 0.03-P3HT/PEDOT:PSS HTL.

**Figure 8 nanomaterials-14-01476-f008:**
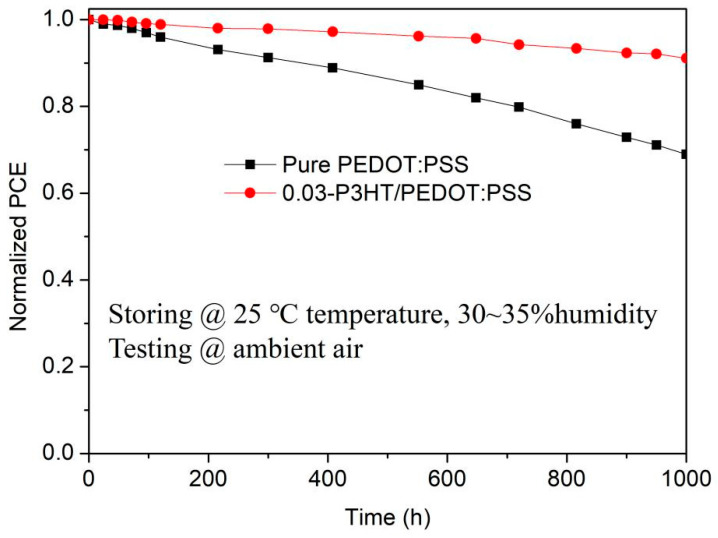
Stability of the PSCs based on the pure PEDOT:PSS HTL and the 0.03-P3HT/PEDOT:PSS HTL. The encapsulated PSCs were stored under environmental conditions of 25 °C temperature and 30~35% humidity and tested under ambient air.

**Table 1 nanomaterials-14-01476-t001:** Summary of the photovoltaic performances of the PSCs using the pure PEDOT:PSS HTL and the PEDOT:PSS HTLs incorporating different P3HT concentrations. The data within parentheses indicates the optimal performance of the inverted solar cells.

Device	V_oc_ (V) ^a^	J_sc_ (mA.cm^−2^) ^a^	FF (%) ^a^	PCE (%) ^a^
Pure PEDOT:PSS	1.00 ± 0.01 (1.00)	20.95 ± 0.91 (21.85)	76.11 ± 1.82 (77.98)	16.49 ± 0.6 (17.04)
0.01-P3HT/PEDOT:PSS	1.00 ± 0.01 (1.00)	21.02 ± 0.78 (21.98)	77.67 ± 2.11 (79.90)	16.81 ± 0.89 (17.61)
0.02-P3HT/PEDOT:PSS	1.01 ± 0.01 (1.02)	22.17 ± 0.86 (22.97)	78.21 ± 2.02 (79.97)	17.93 ± 0.92 (18.81)
0.03-P3HT/PEDOT:PSS	1.02 ± 0.01 (1.03)	22.94 ± 0.76 (23.38)	80.31 ± 2.14 (82.14)	18.96 ± 0.63 (19.78)
0.04-P3HT/PEDOT:PSS	1.01 ± 0.01 (1.02)	21.85 ± 0.99 (22.45)	78.21 ± 3.21 (80.09)	17.84 ± 0.71 (18.34)

^a^ The averaged PCEs were computed from a total of twelve solar cells.

## Data Availability

The data will be made available upon request.

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
