# Peer review of "Simultaneously Enhancing the Efficiency and Stability of Perovskite Solar Cells by Using P3HT/PEDOT:PSS as a Double Hole Transport Layer"

_nanomaterials, 2024, doi:10.3390/nano14181476_

Round 1
Reviewer 1 Report
Comments and Suggestions for Authors
The paper investigates the use of P3HT/PEDOT:PSS as a double hole transport layer to enhance the efficiency and stability of perovskite solar cells. However, the idea presented in this work is not particularly new. Currently, the power conversion efficiency (PCE) of perovskite solar cells has already reached approximately 26%, so the authors should aim to achieve a PCE of at least 23% to make a more significant contribution.
Additionally, there are some small issues with the manuscript that need to be addressed:
- Figure: The figure numbers in the manuscript are incorrect. The description of Figure 8, in particular, needs to be revised for clarity to ensure that other readers can easily understand the content.
- Stability Test Conditions: The conditions under which the stability tests were conducted are not provided. This information is crucial for evaluating the reliability of the stability results and should be included in the manuscript.
Author Response
The paper investigates the use of P3HT/PEDOT:PSS as a double hole transport layer to enhance the efficiency and stability of perovskite solar cells. However, the idea presented in this work is not particularly new. Currently, the power conversion efficiency (PCE) of perovskite solar cells has already reached approximately 26%, so the authors should aim to achieve a PCE of at least 23% to make a more significant contribution.
Response: Thanks for the comment. PCE of perovskite solar cells (PCSs) based on PEDOT:PSS hole transport layer (HTL) has been optimized in our laboratory. However, the baseline PCE of PCSs has not been improved yet. In this work, we focused on studying the P3HT-modified PEDOT: PSS HTL and observed the trend of changes after P3HT modification at the same baseline efficiency. We will further improve our baseline PCE in the future.
Additionally, there are some small issues with the manuscript that need to be addressed:
Figure: The figure numbers in the manuscript are incorrect. The description of Figure 8, in particular, needs to be revised for clarity to ensure that other readers can easily understand the content.
Stability Test Conditions: The conditions under which the stability tests were conducted are not provided. This information is crucial for evaluating the reliability of the stability results and should be included in the manuscript.
Response: Thanks for your nice reminder. We have corrected the incorrect figure numbers in the revision. In addition, the conditions under which the stability tests have been added in the experimental section in the revision. The description of Figure 8 has been revised in the revision to make it easier to understand by other readers.
Reviewer 2 Report
Comments and Suggestions for Authors
This work explores the design of highly efficient and stable perovskite solar cells through interface engineering using a highly hydrophobic P3HT layer positioned between the conventional HTL (PEDOT:PSS) and the perovskite layer. The incorporation of P3HT enhances the size and crystallinity of perovskite grains, potentially leading to an increased efficiency. Additionally, the hydrophobic nature of P3HT may contribute to improved stability. However, this manuscript contains several flaws and requires a significant revision (major revision) before it can be considered for publication in “Nanomaterials”.
1. A comparison table should be included for summarizing and comparing the present work with the literature reports on the use of hydrophobic polymers in the conventional HTL (PEDOT:PSS) of perovskite solar cells.
2. Instead of figure numbers in manuscript text, they were labelled with a '0' symbol. It must be modified clearly in the revised manuscript.
3. The scale in Fig. 1b is confusing. It needs to be rechecked. The concentration of P3HT on the X-axis (0.1-0.8) differs from the concentration mentioned in the text (0.05 mg/mL). Please ensure that the unit should also be included on the X-axis.
4. The authors should include the contact angle measurements of the optimized film (0.03 mg/mL) with water and perovskite solution in the main manuscript, moving other concentration data to Supporting Information.
5. The figure caption of Fig. 3 appears to be incorrect. It should be revised to match the corresponding text in the manuscript.
6. To support the statement "The enhancement in absorption may be due to the contribution of P3HT absorption," the authors should include the absorption spectra of bare P3HT in Supporting Information.
7. Although the authors claim that the inclusion of P3HT between PEDOT:PSS and perovskite improves crystallinity and increases the perovskite grain size, there is no significant enhancement in Voc. The authors should explain the appropriate reason.
8. The statement regarding "improved charge transport capability" could be better supported by including the EIS analysis.
9. Captions for Figures 7 and 8 are improperly labelled and should be revised.
10. While the title mentions a double hole transport layer, the authors need to clearly explain the hole extraction process by illustrating the relevant energy levels.
11. The manuscript contains many typographical errors (e.g., PDDOT, Deviece) and that should be rechecked and corrected in the whole manuscript.
Comments on the Quality of English LanguageMinor editing of English language required.
Author Response
This work explores the design of highly efficient and stable perovskite solar cells through interface engineering using a highly hydrophobic P3HT layer positioned between the conventional HTL (PEDOT:PSS) and the perovskite layer. The incorporation of P3HT enhances the size and crystallinity of perovskite grains, potentially leading to an increased efficiency. Additionally, the hydrophobic nature of P3HT may contribute to improved stability. However, this manuscript contains several flaws and requires a significant revision (major revision) before it can be considered for publication in “Nanomaterials”.
Response: Thanks for the high rating of our work. We have carefully answered your questions point by point as below.
1. A comparison table should be included for summarizing and comparing the present work with the literature reports on the use of hydrophobic polymers in the conventional HTL (PEDOT:PSS) of perovskite solar cells.
Response: Thank you very much for the valuable comments. Based on the reviewer's suggestion, we have re-evaluated the efficiency of hydrophobic polymers in the conventional HTL of perovskite solar cells. The PCE of several representative works is listed in Table 1. These studies show that using hydrophobic polymers can significantly enhance the efficiency of perovskite solar cells with conventional PEDOT:PSS as the hole transport layer. These improvements are mainly attributed to the protective effect of hydrophobic polymers on the perovskite layer and the modification of PEDOT:PSS, which reduces interface defects and the impact of moisture.
Table 1 Summarizes and compares the PCE of hydrophobic polymers in the conventional HTLs of perovskite solar cells.
|
Device |
hydrophobic polymers |
PCE |
Years |
|
ITO/poly TPD/Perovskite/PC61BM/C60/BCP/Au |
Poly TPD |
15.15%[1] |
2019 |
|
ITO/PEDOT:PSS/Perovskite/PC60BM/Ag |
PTAA |
16.94%[2] |
2019 |
|
ITO/PEDOT:PSS/PTAA/Perovskite/PC60BM/Ag |
19.04%[2] |
||
|
ITO/PTAA/Perovskite/PC60BM/BCP/Ag |
PTAA |
6.90%[3] |
2021 |
|
ITO/PEDOT:PSS/Perovskite/PC60BM/BCP/Ag |
10.10%[3] |
||
|
ITO/PTAA/Perovskite/C60/BCP/Ag |
PTAA:PASQ-IDT |
19.74%[4] |
2022 |
|
ITO/PTAA:PASQ-IDT/Perovskite/C60/BCP/Ag |
21.33%[4] |
||
|
ITO/PTAA/Perovskite/PCBM/BCP/Ag |
PEO-TPAB |
18.33%[5] |
2023 |
|
ITO/PTAA/PEO/Perovskite/PCBM/BCP/Ag |
20.11%[5] |
||
|
ITO/PTAA/PEO-TPAB/Perovskite/PCBM/BCP/Ag |
21.62% [5] |
||
|
ITO/PTAA(PhMe)/Perovskite/C60/BCP/Ag |
PTAA(PhMe) and PTAA(Py) |
19.35%[6] |
2023 |
|
ITO/PTAA(Py)/Perovskite/C60/BCP/Ag |
20.53%[6] |
||
|
ITO/PEDOT:PSS/Perovskite/PCBM/BCP/Ag |
P3HT |
17.04% |
This work |
|
ITO/PEDOT:PSS/P3HT/Perovskite/PCBM/BCP/Ag |
19.78% |
- J Höcker, D Kiermasch, P Rieder, K Tvingstedt, A Baumann, V Dyakonov. Efficient Solution Processed CH3NH3PbI3 Perovskite Solar Cells with PolyTPD Hole Transport Layer[J]. Z. Naturforschung A. 74(2019) 665-672. https://doi.org/10.1515/zna-2019-0127.
- M Wang, H Wang, W Li, X Hu, K Sun, Z Zang. Defect Passivation Using Ultrathin PTAA Layers for Efficient and Stable Perovskite Solar Cells with a High Fill Factor and Eliminated Hysteresis[J]. J. Mater. Chem. A. 7(2019) 26421-26428. https://doi.org/10.1039/C9TA08314F.
- H Mehdi, M Matheron, A Mhamdi, S Cros, A Bouazizi. Effect of the Hole Transporting Layers on the Inverted Perovskite Solar Cells[J]. J Mater Sci Mater Electron. 32(2021) 21579-21589. https://doi.org/10.1007/s10854-021-06666-z.
- F Wu, Q Xiao, X Sun, T Wu, Y Hua, L Zhu. Hole Transporting Layer Engineering Via a Zwitterionic Polysquaraine Toward Efficient Inverted Perovskite Solar Cells[J]. Chem. Eng. J. 445 (2022): 136760. https://doi.org/10.1016/j.cej.2022.136760.
- J Dai, J Xiong, N Liu, Z He, Y Zhang, S Zhan, B Fan, W Liu, X Huang, X Hu, D Wang, Y Huang, Z Zhang, J Zhang. Synergistic Dual-interface Modification Strategy for Highly Reproducible and Efficient PTAA-based Inverted Perovskite Solar Cells[J]. Chem. Eng. J. 453 (2023): 139988. https://doi.org/10.1016/j.cej.2022.139988.
- Y Wang, J Song, L Chu, Y Zang, Y Tu, J Ye, Y Jin, G Li, Z Li, W Yan. Buried Solvent Assisted Perovskite Crystallization for Efficient and Stable Inverted Solar Cells[J]. J. Power Sources. 558 (2023): 232626. https://doi.org/10.1016/j.jpowsour.2023.232626.
2. Instead of figure numbers in manuscript text, they were labelled with a '0' symbol. It must be modified clearly in the revised manuscript.
Response: Thanks for your kind reminder. We have carefully revised figure numbers for clarity and ease of understanding.
3. The scale in Fig. 1b is confusing. It needs to be rechecked. The concentration of P3HT on the X-axis (0.1-0.8) differs from the concentration mentioned in the text (0.05 mg/mL). Please ensure that the unit should also be included on the X-axis.
Response: Thanks for the comment. We measure contact angle (CA) of PEDOT:PSS modified with different P3HT concentration, ranging from 0.01 to 0.8 mg/mL in our work. It can be seen the CA significantly increase with the concentration of P3HT increase when P3HT concentration below 0.1 mg/mL. However, the CA slowly increase with the concentration of P3HT further increase when P3HT concentration is greater than 0.1 mg/mL. In our work, it has been found that it is very difficult to form uniform perovskite films when P3HT concentration is higher than 0.05 mg/mL. Therefore, various concentration of P3HT including 0.01, 0.02, 0.03, 0.04 mg/mL were studied in below work.
4. The authors should include the contact angle measurements of the optimized film (0.03 mg/mL) with water and perovskite solution in the main manuscript, moving other concentration data to Supporting Information.
Response: Thanks for the comment. The contact angle measurements of the optimized film (0.03 mg/mL) with water and perovskite solution in Fig. 1b. The contact angle measurements of the optimized film (0.03 mg/mL) with water and perovskite solution is 56 and 24 degrees, respectively.
5. The figure caption of Fig. 3 appears to be incorrect. It should be revised to match the corresponding text in the manuscript.
Response: Thanks for your nice reminding. We have carefully corrected the incorrect figure caption to match the corresponding text in the revision.
6. To support the statement "The enhancement in absorption may be due to the contribution of P3HT absorption," the authors should include the absorption spectra of bare P3HT in Supporting Information.
Response: Thanks for your nice suggestion. We have added the references about the absorption spectra of bare P3HT in the revision.
- Kaori Yaguchi, Akihiro Furube, and Ryuzi Katoh. Study of Ultrathin Films of P3HT/PCBM by Means of Highly Sensitive Absorption Spectroscopy. Chem. Lett. 2012, 41, 184-186. https://doi.org/10.1246/cl.2012.184.
- Erkin Zakhidov, Mukhib Imomov, Vakhob Quvondikov, Sherzod Nematov, Ilkhom Tajibaev, Aziz Saparbaev, Irfan Ismail, Bilal Shahid, Renqiang Yang. Comparative study of absorption and photoluminescent properties of organic solar cells based on P3HT:PCBM and P3HT:ITIC blends. Appl. Phys. A, 2019, 125:803. https://doi.org/10.1007/s00339-019-3100-0.
7. Although the authors claim that the inclusion of P3HT between PEDOT:PSS and perovskite improves crystallinity and increases the perovskite grain size, there is no significant enhancement in Voc. The authors should explain the appropriate reason.
Response: Thanks for the comment. Our analysis of AFM, SEM and XRD test results indicates that the introduction of P3HT improves the crystallization of perovskite film and perovskite grain size. However, perovskite grain size did not increase significantly. Therefor, there is no significant enhancement in Voc in our perovskite solar cells.
8. The statement regarding "improved charge transport capability" could be better supported by including the EIS analysis.
Response: Thanks for your nice suggestion. All results, including the external quantum efficiency (EQE), photoluminescence (PL), and conductivity of the PSCs confirm that the introduction of P3HT improves charge transport capability, as shown in Fig. 6b and Figa. 7a-b. In the future, we plan to delve into the charge transfer mechanism through EIS measurements. This will allow us to better understand how charges are transferred in the system.
9. Captions for Figures 7 and 8 are improperly labelled and should be revised.
Response: Thanks for your nice reminding. We have carefully corrected captions in the revision to make it easier to understand by other readers.
10. While the title mentions a double hole transport layer, the authors need to clearly explain the hole extraction process by illustrating the relevant energy levels.
Response: Thanks for your nice suggestion. We have carefully explained the hole by illustrating the relevant energy levels in the revision.
11. The manuscript contains many typographical errors (e.g., PDDOT, Deviece) and that should be rechecked and corrected in the whole manuscript.
Response: Thanks for your kind reminder. We have carefully revised typographical errors in the whole text to make it clearer and easier to understand.
Reviewer 3 Report
Comments and Suggestions for Authors
The authors studied the effect of bilayer hole transport layer containing PEDOT:PSS and P3HT and the focus was the effect of different P3HT concentrations on the perovskite films and then the performance & lifetime of perovskite solar cells. Before considering publication, the whole structure of the manuscript and the gramma & typo do need to be improved.
1. In part 3.3, the authors discussed absorption of the perovskite films on PEDOT:PSS and x-P3HT/PEDOT:PSS. The wavelength of the absorption spectra was between 500 – 800 – 850 nm, which is much narrower than a solar spectrum. In order to match the EQE spectra in a wavelength range of 300 – 800 nm, the authors need to show the similar range of absorption spectra. Also, although the authors mentioned in line 173 – 176 about the absorption ability, no corresponding discussion continued in the following device performance parts.
2. In part 3.4, the authors used XRD to prove that P3HT inhibited the decomposition of perovskite films. Is it an immediate effect or the long-lasting one? The authors need to clarify the experimental details about the measurement.
3. In part 3.5, The authors discussed the device performance including PV parameters and EQE spectra. When it comes to EQE, the authors attributed the higher EQE to the improved charge transport capability, and then showed figure 7a to prove the higher conductivity of 0.03-P3HT/PEDOT:PSS HTL. I have a concern about the way the authors used to prove the higher conductivity. The structure of ITO/HTL/Ag is used for hole transport, although the slope of 0.03-P3HT/PEDOT:PSS is higher, it shows as a resistor. Any possibility the higher slope is causing higher leaky current but not due to higher conductivity?
4. The authors need to clarify how the lifetime measurement was done. Is it a shelf lifetime or a photo-stressing measurement?
5. All the figures and table need to mark the number.
6. All the abbreviations need to have the full name at the first time showing in the text.
Comments on the Quality of English Language
7. Please carefully check the gramma and typos in the whole text. For example, line 21, 90-91, 96, 129, and 175 etc.
Author Response
The authors studied the effect of bilayer hole transport layer containing PEDOT:PSS and P3HT and the focus was the effect of different P3HT concentrations on the perovskite films and then the performance & lifetime of perovskite solar cells. Before considering publication, the whole structure of the manuscript and the gramma & typo do need to be improved.
Response: Thanks for the high rating of our work. We have carefully answered your questions point by point as below.
- In part 3.3, the authors discussed absorption of the perovskite films on PEDOT:PSS and x-P3HT/PEDOT:PSS. The wavelength of the absorption spectra was between 500 – 800 – 850 nm, which is much narrower than a solar spectrum. In order to match the EQE spectra in a wavelength range of 300 – 800 nm, the authors need to show the similar range of absorption spectra. Also, although the authors mentioned in line 173 – 176 about the absorption ability, no corresponding discussion continued in the following device performance parts.
Response: Thanks for the comment. We present absorption spectra within a wavelength range of 300~800 nm in the revision. It can be seen that all perovskite thin films show almost consistent absorption in the wavelength range of 600~850 nm and 300~490 nm, while the absorption ability of perovskite film with P3HT is significantly higher than that of perovskite film with pure PEDOT:PSS in the wavelength range of 490~600 nm. The enhancement in absorption may be due to the contribution of P3HT absorption, which is stronger in the wavelength range of 490~600 nm. Therefore, we believe that the absorption of perovskite film has hardly changed, and the enhancement in PCS performance is not from the contribution of absorption.
- In part 3.4, the authors used XRD to prove that P3HT inhibited the decomposition of perovskite films. Is it an immediate effect or the long-lasting one? The authors need to clarify the experimental details about the measurement.
Response: Thanks for the comment. Many studies suggest that the decrease of PbI2 peak for the perovskite film in XRD measurement can confirm the decomposition of perovskite films is effectively inhibited. We find the PbI2 peak of the perovskite film with P3HT is lower than that of the perovskite film with pure PEDOT:PSS in our work. Therefore, P3HT inhibited the decomposition of perovskite films.
- In part 3.5, The authors discussed the device performance including PV parameters and EQE spectra. When it comes to EQE, the authors attributed the higher EQE to the improved charge transport capability, and then showed figure 7a to prove the higher conductivity of 0.03-P3HT/PEDOT:PSS HTL. I have a concern about the way the authors used to prove the higher conductivity. The structure of ITO/HTL/Ag is used for hole transport, although the slope of 0.03-P3HT/PEDOT:PSS is higher, it shows as a resistor. Any possibility the higher slope is causing higher leaky current but not due to higher conductivity?
Response: Thanks for the comment. Indeed the structure of ITO/HTL/Ag is a resistor. However, the slope of the current-voltage curve in Figure 7a comes from the ratio of current to voltage rather than the ratio of voltage to current. Therefore, the slope indicates conductivity, not resistance. Consequently, the higher slope is causing higher conductivity.
- The authors need to clarify how the lifetime measurement was done. Is it a shelf lifetime or a photo-stressing measurement?
Response: Thanks for the comment. The conditions under which the stability tests have been added in the experimental section in the revision. The description of Figure 8 has been revised in the revision to make it easier to understand by other readers.
- All the figures and table need to mark the number.
Response: Thanks for your kind reminder. We have marked the number in all figures and table in the revision.
- All the abbreviations need to have the full name at the first time showing in the text.
Response: Thanks for your kind reminder. We have provided full names for all abbreviations in the revision to make it easier for readers to understand.
- Please carefully check the gramma and typos in the whole text. For example, line 21, 90-91, 96, 129, and 175 etc.
Response: Thanks for your kind reminder. We have carefully revised the gramma and typos in the whole text to make it clearer and easier to understand.
Round 2
Reviewer 1 Report
Comments and Suggestions for Authors
Good to publish
Author Response
Thank you for your review and recommendations.
Reviewer 2 Report
Comments and Suggestions for Authors
Although the authors tried to address some issues arisen from the 1st review round, the paper still lacks critical points that should be clarified before acceptance for publication in Nanomaterials.
1. Please add the comparison table in the main manuscript or provide it as Supporting Information.
2. Figure numbers are still labelled by '0' in the revised manuscript. The authors should be careful and diligent for their revision.
3. In Fig. 1(c), there is no unit of ‘Concentration of P3HT’ on the x-axis.
4. In relation to Comment #4 at the 1st review round, the authors should include information about the contact angles in the manuscript text.
5. In relation to Comment #6 at the 1st review round, the authors should provide their experimental results (absorption spectra) from the actual materials used in this work. Only citing some references from other researchers is inadequate to support the authors’ claim.
6. In relation to Comment #7 at the 1st review round, the magnitude of Voc is related to not only the grain size but also to the crystallinity. The authors should re-interpret the appropriate reason why there is no significant enhancement in Voc in spite of the improved crystallinity in P3HT-incorporated samples.
7. In relation to Comment #7 at the 1st review round, the authors answered “We have carefully explained the hole by illustrating the relevant energy levels in the revision.”. However, in the revised manuscript, there is no energy band scheme and its related interpretation. Please provide the energy band diagram of the proposed device, and explain the appropriate transport mechanism. Particularly, compare and explain two different device types with ans without P3HT.
8. According to iThenticate reports, the present form of the manuscript still includes 35% of plagiarism. Even though it seems not intentional, the authors need to check and rephrase the related sentences.
Comments on the Quality of English LanguageAccording to iThenticate reports, the present form of the manuscript still includes 35% of plagiarism. Even though it seems not intentional, the authors need to check and rephrase the related sentences.
Author Response
Although the authors tried to address some issues arisen from the 1st review round, the paper still lacks critical points that should be clarified before acceptance for publication in Nanomaterials.
Response: Thanks for the comment. We have carefully answered your questions point by point as below.
- Please add the comparison table in the main manuscript or provide it as Supporting Information.
Response: Thanks for your nice suggestion. We have added the comparison table to the Supporting Information, available in Table S1.
- Figure numbers are still labelled by '0' in the revised manuscript. The authors should be careful and diligent for their revision.
Response: Thanks for your kind reminder. We have carefully revised figure numbers for clarity and ease of understanding.
- In 1(c), there is no unit of ‘Concentration of P3HT’ on the x-axis.
Response: Thanks for your nice reminding. The unit of ‘Concentration of P3HT’ have been added to Fig. 1(c).
- In relation to Comment #4 at the 1st review round, the authors should include information about the contact angles in the manuscript text.
Response: Thanks for your nice reminding. The information about the contact angles has been added to the manuscript text.
- In relation to Comment #6 at the 1st review round, the authors should provide their experimental results (absorption spectra) from the actual materials used in this work. Only citing some references from other researchers is inadequate to support the authors’ claim.
Response: Thanks for your helpful suggestion. We have measured the absorption spectra of P3HT and added them to Supporting Information in Fig. S1.
- In relation to Comment #7 at the 1st review round, the magnitude of Voc is related to not only the grain size but also to the crystallinity. The authors should re-interpret the appropriate reason why there is no significant enhancement in Voc in spite of the improved crystallinity in P3HT-incorporated samples.
Response: Thank you for your comment. According to our measurements, both the grain size and crystallinity of the perovskite have been increased to varying degrees. The Voc has risen from 1.00 V to 1.02 V. The value of Voc depends not only on the grain size and crystallinity but also on other factors such as the contact interface between the interfacial layer and the perovskite film, as well as the electrode. Consequently, we have not observed a significant enhancement in Voc in our perovskite solar cells.
- In relation to Comment #7 at the 1st review round, the authors answered “We have carefully explained the hole by illustrating the relevant energy levels in the revision.”. However, in the revised manuscript, there is no energy band scheme and its related interpretation. Please provide the energy band diagram of the proposed device, and explain the appropriate transport mechanism. Particularly, compare and explain two different device types with ans without P3HT.
Response: Thanks for your valuable suggestion. We have incorporated the energy band diagram of the proposed device into Figure 6. Additionally, the transport mechanism is thoroughly analyzed in the revised document.
According to iThenticate reports, the present form of the manuscript still includes 35% of plagiarism. Even though it seems not intentional, the authors need to check and rephrase the related sentences.
Response: Thanks for your nice suggestion. We have carefully check and rephrase the related sentences.
Reviewer 3 Report
Comments and Suggestions for Authors
The absorption spectra in shorter wavelength is not the real signal. Plesae explain why the saturation is showed. Is it caused by the equipment limitation?
With regards to the stablity, the author still need to make it clear if the deivces were stored in the dark and only illuminated when testing IV.
Comments on the Quality of English LanguageLine 110, it is a complete sentence "Taking PEDOT:PSS for filtration, and then diluting the filtered PEDOT:PSS with deionized water in a 1:4 ratio."
Author Response
The absorption spectra in shorter wavelength is not the real signal. Please explain why the saturation is showed. Is it caused by the equipment limitation?
Response: Thanks for comment. Yes, the saturation phenomenon is usually due to equipment limitations.
With regards to the stability, the author still need to make it clear if the devices were stored in the dark and only illuminated when testing IV.
Response: Thanks for comment. All the devices were stored in the dark and only illuminated when testing I-V. We have rephrase the related sentences in the revision to make them easier to understand.
Line 110, it is a complete sentence "Taking PEDOT:PSS for filtration, and then diluting the filtered PEDOT:PSS with deionized water in a 1:4 ratio."
Response: Thank you for your suggestion, we have corrected this sentence. "Filter PEDOT:PSS, and then dilute the filtered PEDOT:PSS with deionized water in a 1:4 ratio."
Round 3
Reviewer 2 Report
Comments and Suggestions for Authors
The authors addressed all the comments and concerns arisen from the previous review rounds. Thus, I recommend this paper for publication in Nanomaterials.
Comments on the Quality of English LanguageMinor editing of English language required.